# What does it mean to have experienced the death of a relative in a context of social and funeral restrictions? Lessons from the pandemic for bereavement research and clinical practice

Camille Boever[1,2]*, Emmanuelle Zech[1], Laurence Arcand[3], Chantal Verdon[3]

1 Psychological Sciences Research Institute of the Université catholique de Louvain (UCLouvain), Louvain-la-Neuve, Belgium, 2 Fund for Scientific Research (FRS-FNRS), Brussels, Belgium, 3 Department of Nursing, Université du Québec en Outaouais (UQO) – Saint-Jérôme, Québec, Canada

* camille.boever@uclouvain.be

## Abstract

The circumstances of the Covid-19 pandemic raised concerns about their impact on people who were bereaved in this period. Numerous studies have attempted to quantify this impact. However, they have often adopted a pathologizing perspective on grief, with little attention being paid to the mediating processes and to the diversity of experiences, and their results have appeared contradictory. This study takes a comprehensive approach to understand these experiences in their uniqueness, without pathologizing them. Interviews with 12 bereaved people were analyzed using an interpretative phenomenological analysis and drawing on the theoretical framework of meaning-making, to explore the meanings that they gave to their loss and grief in a context of social and funeral restrictions. In our findings, the context of the pandemic appeared to be completely incompatible with the field of dying and mourning, creating paradoxical injunctions for some participants. While this led to feelings of guilt, powerlessness and loss of meaning for some, others were able to experience meaningful moments and to find solace in the farewell. All experiences were far from what had been expected, for better or for worse, and participants had to find their own ways to make sense of these unexpected experiences. The meanings they gave were complex, combining different levels of meaning – personal, moral, societal, or existential – and evolved over time, as did their emotional experiences. The results highlight the notions of paradoxical injunctions and grieving for an ideal goodbye as relevant for understanding and supporting the bereaved, drawing avenues for therapeutic work (e.g., restoring agency by providing a secure space to create a meaningful rite). They also have implications for research, highlighting the need to broaden the understanding of "impact", to include mediators assessing subjectivity, and to privilege person-centered qualitative and quantitative methods.

**Data availability statement:** In accordance with Data Protection principles, the research data is not available online due to ethical restrictions, as it includes sensitive and clinical information that could potentially identify the participants. Moreover, the Ethics Committee requires that interview data not be disclosed, only its analysis and carefully selected verbatims to illustrate findings. However, the research data may be accessible from the corresponding author upon reasonable request and subject to the approval of the Ethics Committee. Requests can be sent to the corresponding author, or to the body responsible for research administration at UCLouvain (email: privacy@uclouvain.be) specifying the names of the authors. If the request is accepted by the Ethics Committee, a Data Transfer Agreement form will be completed, as required by the researchers' institution (UCLouvain).

**Funding:** CB was supported by a Grant "Fonds pour la Recherche en Sciences Humaines" (FRESH) from the "Fonds de la recherche Scientifique" (F.N.R.S.-F.R.S.) [number 1.F.010.22F.] website : https://www.frs-fnrs.be/fr/financements/chercheur-doctorant#fresh They did not play any role in the study design or any step of the research.

**Competing interests:** The authors have declared that no competing interests exist.

## Introduction

This paper explores the experiences of grief following the loss of a close one during the pandemic, when funeral rites and social gatherings were restricted, and draws lessons for clinical and bereavement research. Grief refers to the set of responses to the experience of loss. In this article, this notion encompasses dynamic, ongoing, intrapersonal and interpersonal processes of adjusting to and integrating loss, as well as the emotional experience of this loss, giving rise to evolving psychological, physical, social, behavioral and existential reactions that are specific to each person [1,2]. While the concept and experience of grief can be applied more broadly to any type of loss [2], in this study, we limit our definition to the death of a close one.

In 2020, restrictions aimed at stopping the Covid-19 viral contamination have limited access to places of worship and care, and restricted social gatherings as well as funeral and mourning rites. Deploying historical, psychological or socio-anthropological arguments on the importance of funeral rites in adjustment to loss [3,4], many authors anticipated that the impossibility of such rites, combined with other risk factors, would hinder grief processes, prevent mourners from realizing and making sense of the loss, from receiving social support [5,6], and ultimately increase pathological grief reactions [7,8].Other authors [9,10] called for caution in predicting psychopathological outcomes, arguing that the performance of funeral rites (or their absence) does not show a consistent link with the intensity of grief reactions [11], and that their meaning evolves and differs from person to person [12,13].

Since then, numerous studies have addressed the initial and urgent questions about the impact of restrictions on grief. This has mainly been done from two angles: either by assessing grief reactions or by identifying the specific circumstances that affected the bereaved. First, authors have attempted to identify the consequences of losing a relative in these circumstances (for a review, see [14]), by estimating prevalences of diagnoses of pathological grief [15,16] or assessing the intensity of grief reactions [17,18]. However, due to differences in methodology, their results are difficult to compare, making it hard to determine the effect of circumstances on grief reactions and the nature, extent, and meaning of that effect. Some qualitative studies have revealed that restrictions have been experienced differently by individuals, sometimes leading to feelings of relief, opportunities to rethink or create meaningful ways of accompanying and saying goodbye to the deceased, or to grieve more freely [19,20]. Yet the main narrative that has emerged portrays circumstances as having a strong, and mostly negative impact on opportunities to say goodbye, rites, and the subsequent grieving processes [21,22], provoking feelings of disbelief about the death [23], powerlessness [19], guilt towards the deceased [24], or a lack of recognition of grief [25]. Secondly, studies have sought to identify the individual, relational and situational factors in the context of the pandemic that affected bereaved people or exacerbated their distress [26,27]. Qualitative studies have revealed that the impossibility of accompanying the deceased at the end of life, the accumulation of other stressors and losses, social isolation, or the inadequacy of restricted in-person or virtual funeral rites particularly affected bereaved people [21,28]. However, when

the impact of these factors on grief reactions has been empirically tested, contradictions emerged: studies that have examined whether greater restrictions at the time of the loss were associated with more intensity of grief have yielded mixed results [29,30], as have those that tested whether distress increased when rites were more restricted [18,26] or less satisfying [31,32].

Those studies have partially described what bereaved people experienced during this period, outlined the possible consequences for their grief and suggested ways to support them therapeutically. However, the adoption of these two angles had several blind spots. First, they took a psychopathological perspective on grief: studies that have quantitatively examined grief during the pandemic have primarily looked for 'symptoms' or diagnoses of 'pathological grief' [26,33] (for a historical and clinical critique of the notion of pathological grief, see [34]). Some qualitative studies have also described the experiences in terms of particularly intense grief reactions [21,28]. This relatively reductive approach to bereavement affects our ability to understand grief as a holistic, complex and multidimensional experience [2]. It leads researchers to look for psycho(patho)logical reactions, rather than exploring the experiential reality of the bereaved, whether distressed or not, thus reducing the possibility of observing diverse experiences. Yet, not only did circumstances vary enormously during the pandemic, but it is also known that even in similar circumstances, the ways people experience the death of someone close, find meaningful ways of saying goodbye, and grieve, are highly subjective, idiosyncratic, and inextricably linked to the meanings they give to it [35,36]. One way of exploring this diversity of experiences is thus to acknowledge their singularity, by studying mediating processes of meanings, such as how people appraise the loss, interpret it and what sense they do make of it. Some studies took this perspective and showed how experiences of loss under restrictions prevented some people from finding closure [23], were difficult to integrate and to give meaning to [14], and led to incoherent narratives [37]. Other studies showed that some bereaved people were able to make sense of their experiences, by seeking comfort, gratitude [38], social connection, coherence and recognition [39], and that giving meaning allowed them to partially mitigate the effects of funeral restrictions on their grief [29].

Finally, most authors discussed their findings in the very specific context of the pandemic. It is true that pandemic grief appears to have unique characteristics [25], and that studying it can help us better understand what people experienced, provide appropriate support, and anticipate decisions and actions in similar future contexts [40]. We believe that the study of bereavement in an unprecedented context of changes and restrictions in the conditions of end-of-life, death and mourning also offers a unique opportunity to improve knowledge in a number of areas, namely the meanings of funeral rites and the functions they fulfil in bereavement, and the needs of the bereaved and how they react, adjust, create and reimagine when these needs are thwarted, as this can also occur in other circumstances (e.g., when the deceased had opposite wishes for their funeral than the bereaved, or when the close person cannot be present when the beloved person dies, such as in situations of migration or detention). In this way, studies can shed light on the specific context of the pandemic but also draw lessons for research or clinical work on bereavement in general.

The present study builds on these reflections and seeks to explore the experience of bereavement during this period of social and funeral restrictions from a meaning-making perspective. We aim to answer the following question: "*What meanings do bereaved people give to their experience of loss and grieving in circumstances of restricted funeral rites and social gatherings?*" by analyzing 12 interviews conducted with bereaved people using the method of interpretative phenomenological analysis [41]. By adopting a comprehensive approach, we aim to understand the meaning of lived experiences in all their uniqueness, complexity and nuance, while taking care to move away from the limitations mentioned above. First, by seeking the meaning given to experience, we aim to access what is less obvious in language, going beyond a descriptive and literal account of events and reactions to understand intimate and subjective experiences [36]. Secondly, this study deliberately aims to explore a diversity of experiences, taking care not to look specifically for negative experiences or to adopt a pathologizing prism. Thirdly, it involves people who lost their relative 1–3 years before. This time lapse allows a temporal perspective that invites the interviewees to relate differently to their experience, and to address what has changed in the account of their experience and in the meaning they give to it. It also allows the researchers to step back

from pressing issues and discuss the results from a more overarching perspective, looking for lessons to be drawn from these experiences for clinical practice and research in the field of bereavement.

## Theoretical framework

This study uses a psychological framework and approaches grief as conceptualized in Bonanno and Kaltman's *Integrative Perspective on Bereavement* [35]. This model considers the experience of grief as composed of different elements: the context of loss, the continuum of subjective meanings, the changing representations of the lost relationship, and the set of coping and emotional regulation processes. The model presents its components as interrelated, evolutive and specific to each person, acknowledging the complexity, singularity and diversity of grieving processes and trajectories. In this study, we focus our attention on the subjective meanings that the person gives to the experience of loss. As in the model, we refer to them as a continuum, encompassing subjective perceptions of circumstances and consequences of the loss, interpretations of the cause and meaning of death, and more existential questions about the bereaved's own identity, the place of the deceased or the meaning of life. However, we also acknowledge the importance of the other aspects of grieving experiences. Firstly, the context of loss, which we interpret in a broad sense, including the characteristics of the bereaved person, the deceased person, the circumstances of the death, the person's social, financial and cultural resources, and the socio-cultural context in which the loss and bereavement take place [42]. Secondly, the changing representations of the relationship, including the redefinitions, relocations and transformations of the bond with the deceased and the meaning given to this bond, inspired by Klass et al.'s theory of *continuing bonds* [43,44]. Finally, the coping and emotional regulation processes through which, consciously or unconsciously, the bereaved attempt to cope with the myriads of negative and positive emotions aroused by the loss, its consequences and the relational and existential questions it engenders.

The present study is also grounded in the theoretical framework of meaning-making, developed by Park [45] and inspired by the work of other authors [46,47] and that specifically addresses how people make meaning of their experiences. This model posits that each person possesses a global meaning system that includes a set of beliefs, internal representations, values, feelings, knowledge, and expectations about existence, the world, and oneself, through which he or she consciously or unconsciously understands and makes sense of experiences. It postulates that when the situational meaning given to a lived experience diverges from the person's global meaning system and core beliefs, this can cause distress and initiate a process of meaning reconstruction aimed at reducing the perceived incongruence [45]. This meaning-making process can take different forms [45,48], involve multiple levels of experience, ranging from practical, relational, and identity-related aspects to more existential aspects related to the meaning of life or one's role in it [49], and lead to the generation of a multitude of meanings. In our study, we will consider meaning from a broad perspective, including the processes by which the person attempts to make meaning, the meanings that are formulated, and their evolution, as recommended [45].

## Methodology

To explore the meanings people gave to their experiences of bereavement in restrictive circumstances, 12 interviews with bereaved people were carried out and then analyzed using the method of interpretative phenomenological analysis (IPA [41]).

These interviews were conducted as part of a longitudinal and mixed-method study, conducted in Belgium and designed to explore the experiences of bereavement after the death of a closed one in a period of social and funeral restrictions. In this study, people who had lost a relative between March 2020 and January 2022 were invited to complete a questionnaire about the death of their relative and about their grief on three occasions, in February 2022, November 2022 and August 2023. Twelve of these people were interviewed face-to-face following the first questionnaire. This study is part of an international collaboration in which similar studies are being conducted [30,50,51] and joint analysis projects are being pursued [52].

## Recruitment, selection and participants

The opportunity to be interviewed was offered to all 472 people who responded to the first questionnaire of the study, whose inclusion criteria were: to be over 18 years of age, to understand French, and to have experienced the death of a significant person in Belgium between March 2020 and January 2022. Recruitment was conducted from 1/02/2022–1/06/2022, through social media, the media, newspaper or association newsletters, word of mouth, and direct personal contact with bereaved people who were mentioned in Belgian digital obituaries. At the end of the questionnaire, 260 people (55.1%) expressed interest in being interviewed by the researcher. A purposive selection method was then defined to select a limited number of people from this group who would cover a wide range of experiences.

On the basis of their answers to the questionnaire, three binary criteria were defined. The first criterion targeted the intensity of grief and aimed at identifying people experiencing various levels of distress. This criterion, based on the total score in the *Traumatic Grief Inventory* (TGI-SR [53]), distinguished people whose score was above or below the cut-off defined by the authors (the theoretical threshold at which clinical concerns about so-called "prolonged" grief may appear [54]). The second criterion focused on subjective satisfaction with the *pre-, peri-* and *post-mortem* rites, distinguishing between those who were fairly or very satisfied and those who were not very or not at all satisfied with the rites carried out for their relative. The third criterion concerned the practice of alternative rites, invented or adapted because of the restrictions at the time of death, and distinguished between those who had answered "*yes*" or "*no*" to the question "*Did you participate in other forms of rites before or after the death of your relative?*". By combining these three criteria, it was possible to identify 9 profiles of experiences, illustrated in Table 1. A total of 18 people were contacted, varying in gender, age, kinship to the deceased, and cause of death. Whenever possible, the researchers tried to meet at least one person per experience profile. Fifteen people responded positively and 12 were met (3 did not follow up), covering 8 of the 9 experience profiles.

Of the 12 interviewees, 5 were men and 7 were women. Their age ranged between 22 and 83 years old. They talked about the death of different relatives (a child, a parent, a spouse, a grandparent, a friend), who had died at different ages (from 8 to 101) and from different causes (illness, accident, suicide). Table 2 shows socio-demographic information for each respondent, as well as some descriptive details about the deceased, the death and their kinship.

## Ethical considerations

The complete study was approved by the Ethical Committee of the Cliniques universitaires Saint-Luc - UCLouvain (#2021/16NOV/483). Only the principal investigator, CB, had access to the identity of the participants and anonymized each transcript. In the results, participants are referred to by a pseudonym, allowing the reader to link the extracts to them. Additional information regarding the ethical, cultural, and scientific considerations specific to inclusivity in global research is included in the Supporting Information (S1 Checklist).

**Table 1. Profiles of experiences by combining the three binary criteria.**

|  | Criteria A (n) | Criteria B (n) | Criteria C (n) |
|---|---|---|---|
| People having accepted the interview N = 260 | TGI > 61 (96) | Satisfying rites (47) | Having created (10) |
|  |  |  | Not having created (37) |
|  |  | Unsatisfying rites (49) | Having created (26) |
|  |  |  | Not having created (22) |
|  | TGI < 61 (164) | Satisfying rites (80) | Having created (33) |
|  |  |  | Not having created (47) |
|  |  | Unsatisfying rites (84) | Having created (32) |
|  |  |  | Not having created (52) |

**Table 2. Socio-demographic data and information concerning the respondent, the deceased, the death and the conduct of the interview.**

| Respondents Name[a], gender[b] (age range) | The deceased Kinship, gender[b] (age range) | Cause of death | Time since death at time of the interview | Context of the interview[c] | Length of the interview |
|---|---|---|---|---|---|
| André, M (60-70) | His partner, W (50-60) | Suicide | 28 months | Research institute | 1h55 |
| Dominique, M (70–80) | His wife, W (60-70) | Cancer | 19 months | Research institute (C + S) | 1h04 |
| Élise, W (30-40) | Her son, M (0-10) | Accident | 29 months | Participant's home (C + S) | 2h24 |
| Henri, M (80-90) | His sister, W (80-90) | Cardiac arrest | 31 months | Participant's home | 1h33 |
| Isabelle, W (60-70) | Her father, M (80-90) | Euthanasia | 21 months | Participant's home | 2h33 |
| Jeanne, W (20-30) | Her father, M (50-60) | Suicide | 24 months | Research institute | 2h18 |
| Louis, M (40-50) | His mother, W (70-80) | Covid-19 | 31 months | Research institute | 2h05 |
| Maria, W (30-40) | Her friend, M (30-40) | Covid-19 | 23 months | Research institute | 1h36 |
| Naomi, W (20-30) | Her father, M (40-50) | Covid-19 | 20 months | Research institute | 2h11 |
| Philippe, M (40-50) | His grandmother, W (100–110) | Covid-19 | 32 months | Public place | 2h00 |
| Sandra, W (60-70) | Her daughter, W (20-30) | Suicide | 14 months | Participant's home (C + S) | 2h36 |
| Tania, W (30-40) | Her father, M (60-70) | Cardiac arrest | 21 months | Participant's home | 1h28 |

[a]The names are pseudonyms. [b]Gender is specified as W for a woman and M for a man. [c]C + S means that the interview was conducted with the researcher and a trainee.

### Data collection and analysis

The 12 interviews were conducted between July and November 2022, between 14 and 32 months after the death of their relative. The participants were interviewed at home, at the research institute or in a public place, according to their preference. Interviews lasted between 64 and 156 minutes ($M = 123.6$, $SD = 26.7$). All participants were interviewed by a female doctoral researcher (CB), trained as a clinical psychologist and trained in IPA as part of her PhD program. For training reasons, a clinical psychology student was present at 3 interviews. Beforehand, CB had an e-mail exchange with the participants to settle the details of the meeting. She explained the context and reasons for the study, the interview procedure, their rights as participants, the potential risks involved in taking part in an interview about their grief, and resources to turn to if they felt the need for more specific support. Before the interview began, the participants were verbally reminded of this information and signed the written consent form.

The interview followed an interview guide developed by CB and validated by EZ and CV. Its aim was to retrace as fully as possible the story of the relative's death and the interviewee's bereavement, exploring various themes such as relationship with the deceased, the period of the end of life and death, the gatherings, gestures and rites that may or may not have been carried out, how these were experienced, as well as the grief experience and its evolution up to the time of the interview. The interview guide suggested a list of questions and possible follow-up questions that were phrased in an open and exploratory way to let the diversity of experiences and unexpected elements emerge. The interview guide was developed before the researchers decided to include the perspective of meaning-making in the research question. For this reason, the questions in the interview guide did not focus on meaning. However, they aimed to retrace comprehensively the lived experience of the participants, from their own perspective, which is a way to access to latent meaning [55]. Examples of questions are: *Can you tell me what happened to the deceased? How were you able to commemorate the death of your loved one? What was important for you at that time? How did you experience this situation? Today, how do you think back to that time and the way you were able to say goodbye to your loved one? How are you feeling now? If so, how would you say it is different from before? Do you think that what you've experienced has changed your relationship with death, rite or mourning?* CB explored the topics with flexibility, allowing for the spontaneous emergence or deeper exploration of topics not covered in the guide. Most of the time, the participants narrated spontaneously their experiences, and the guide was not followed as such. Follow-up questions were then used to gain a deeper understanding of their

emotional experiences (e.g., *What do you mean by that?*). At the end of the interview, CB offered participants the opportunity to debrief and answered their questions.

The audio recording of each interview was transcribed by a psychology student or by CB using Sonix software, and each transcript was revised by CB to add expressions and non-verbal elements (e.g., silence, emotions, changes in speed or intonation), necessary for IPA. The transcripts were then analyzed using the IPA method. With its focus on capturing the way in which people make sense of their experiences [56], IPA enables the study of subjective and evolving meanings. It combines three dimensions of analysis: descriptive (what is said), linguistic (how it is said) and conceptual (what it means), to address the complexity of existential and relational experiences [41]. The analysis was carried out by CB using Nvivo software, in collaboration with CV and LA, specialists in qualitative research and trained in IPA. They followed the guidelines outlined by Smith and Osborn [41], while considering the criticisms that have been levelled at them [57].

First, CB carried out close readings of each interview several times, to immerse herself in each participant's experience. Second, CB, CV, and LA conducted an analytical reading and coding of three interviews, generating descriptive, linguistic, and conceptual comments for each extract. For each disagreement in the interpretation of the data, they clarified their understanding and reached a consensus on meanings that were more consistent with the analysis and with CB's interpretation, who was most engaged with the data and the context of the participants' life. CB and CV identified convergences and divergences among the comments and developed initial and provisional units of meanings. Third, CB coded the following 9 interviews separately and generated comments reflecting their singular experience. Particular attention was paid to the expression of emotional warmth, blocking and repetition in each of the transcripts in order to stay close to the discourse, in all its novelty, unexpectedness and uniqueness [58]. For each interview, CB compared the generated comments with the previously developed units of meanings, identifying common or different aspects of each transcript in relation to the others, and making the units of meanings evolve with each analysis. CB discussed this evolution with LA. An iterative return to previous interviews was carried out, reading each experience in light of the new meanings found. Fourth, after the 12th interview, CB reduced and refined the set of comments and meanings, adopting a more interpretative reading of experiential and existential aspects [58]. The final set of meanings was then structured to develop themes and subthemes, guided by the research question and the theoretical framework. According to the principle of the hermeneutic circle, CB revisited the interviews to ensure that the final themes accurately represented the lived-experiences. Over the fourth step, the articulation, formulation and meaning of the themes were the subject of several discussions between CB, CV and EZ before finally being validated by all the researchers. Thanks to a reviewer's comment, the themes further evolved during the revision of this manuscript, moving towards a more interpretive perspective. The methodology and results were written in accordance with the COREQ checklist and the guidelines of Nizza et al. [58].

**Epistemological and reflexive positioning**

There is some debate in the literature about IPA and whether it unrealistically combines the philosophical foundations of Husserl's descriptive phenomenology and Heidegger's interpretative phenomenology [57]. The present research has an interpretative aim, even if a descriptive analysis was initially necessary before meaning emerged through interpretation [59]. In the interpretative paradigm, researchers cannot eliminate their preconceptions and theoretical assumptions [60]. On the contrary, these assumptions are *part of* the interpretation a researcher can make of a person's interpretation of his or her own experience [41]. In this sense, the principle of *bracketing* was not applied as a method to distance and suspend the researchers' preconceptions. Rather, it was applied as a method for acknowledging, discussing, and clarifying their preconceptions and assumptions through personal and collective reflexivity [61]. In this way, researchers could be (as far as possible) aware of how their assumptions guided their understanding of people's experiences, rather than letting them influence their interpretation in an untransparent way [57,60]. In this study, the authors' personal and professional experiences (a transparent account of each author's subjective position on the subject is presented in Supporting Information (S1 Positioning)) taught them how diverse the needs

of bereaved people are and how their reactions can vary in similar situations. This influenced their explicit search for diversity and the concern not to assume painful experiences, which guided the recruitment methods, the selection process, and the formulation of the questions.

To ensure rigor, the principal researcher, CB, was involved at every stage, from developing the interview guide to conducting, transcribing and analyzing the interviews. Her comprehensive involvement enabled her to gain in-depth knowledge of the interviews and the participants' life contexts. The co-analysis of certain transcripts, as well as numerous discussions with the co-researchers during the analysis, allowed for a transparent and reflexive triangulation which enhanced findings' credibility [62]. In parallel, CB kept a detailed record of the analyses and themes development, as well as a reflexive journal, noting her personal reflections from the reading, and discussing them where necessary with the co-researchers. These exchanges enabled them to confront their interpretative frameworks, influenced by their respective theoretical and experiential backgrounds [57]. As the analyses were carried out 2 years after the interviews, the participants were not asked for feedback on the results.

Furthermore, as stated in the introduction, the study was based on the theoretical framework of meaning-making. Initially, the aim was to explore the experiences of losing and grieving a relative under restrictions, without specific attention to meaning-making. This angle was chosen after having conducted the interviews, using an inductive approach. First, recent analyses of the same study had suggested the importance of addressing the subjective meaning that bereaved people give to their own experiences to understand them better [63]. Then, by reading the interviews and discussing with the co-researchers, CB realized that the search for meaning had guided her approach during the interviews, as they retraced deeply and openly the narratives of people's experiences from their own perspective [55]. It was then decided to use this theoretical framework as the basis for the analysis and the research question was refined to include meanings. The theoretical framework clarified what was included in '*meanings*' and guided the selection of relevant comments, the formulation and articulation of the themes, and the discussion of findings.

## Results

The analysis revealed three themes and their sub-themes (shown in Table 3), which try to capture the meaning the participants gave to their experiences of death and grief, by combining different dimensions of meanings (meaning-making processes and meanings made). The themes are illustrated by interview extracts.

### Contradictions between pandemic and mourning fields acted as paradoxical injunctions

In the account of half of the participants, it appeared that the experience of the end of life and death of a person under social and funeral restrictions confronted them with contradictions between two opposing fields: on the one hand, that of the social and health restrictions imposed during the pandemic, and on the other, that of the human and social needs at the end of life, the death and bereavement of a relative. These contradictions were revealed in 3 dimensions of the rites: humanity/inhumanity, temporality, and sociability.

### The technical rules that were imposed had to be reconciled with human needs

Through the content of the discourse or in the language the participants used, it emerged that the technicist, hygienist, mechanical and cold vision imposed by the restrictions was opposed to the beauty, humanity and sacredness sought at the end of life and death of their relative. The restrictions determined the rules to be followed during the *pre-, peri-* and *post-mortem* rites: both protocols to be respected (e.g., taking specific routes to the coffin, sitting in chairs designated by taping), and technical and managerial behaviors to be adopted (e.g., timing one's speech, selecting authorized attendees). For Jeanne and Philippe, these rules were opposed to their own needs and perspective of the mourning rites. Philippe expressed his frustration with '*saying goodbye to someone on an aluminium bench in the middle of a suburban*

**Table 3. Themes and subthemes generated from the analysis.**

| |
|---|
| **Contradictions between pandemic and mourning fields acted as paradoxical injunctions** |
| • *The technical rules that were imposed had to be reconciled with human needs* |
| • *How could we make sense of a fragmented rite?* |
| • *Everything was put on hold, although it was now or never* |
| • *It made no sense to carry out social rites without getting together* |
| **Experiences were very different from what had been imagined, for better or for worse** |
| • *A multitude of different expectations: between the ideal goodbye, the collective imagination and prescribed norms* |
| • *A multitude of different possibilities for action: from freedom to powerlessness* |
| • *A multitude of meanings: from the extraordinary goodbye to the stolen goodbye* |
| **Experiences had multiple and evolving meanings** |
| • *"I am at peace now"* |
| • *"I'll be angry all my life"* |
| • *"This is death for society? It's horrible, horrible, horrible"* |
| • *"I don't prefer my life now, but I prefer my vision of life"* |

*cemetery'*, although he, like Jeanne, needed beauty and solemnity in such moments. Jeanne described her father's cremation under restrictions as "*a dystopia*", "*violent in its inhumanity*":

> I think I was very, very touched by beauty. (...) And then we arrived in a place that was already ugly (laughs) – really ugly - and not at all personal. And so, we arrived... Oh yes it was really, (...) an ugly, ugly room. They'd put tape on some of the seats, almost police tape, marking the distances we had to keep. (…) So at one point my aunt wanted to come and leave a flower and she didn't follow the right arrow and suddenly [the officer screamed] "NOT OVER THERE" (laughs). We were all there ooooh. It was really... we did not know whether we should laugh or cry, we weren't sure.

In her discourse, the symbolic and behavioral fields related to the pandemic and to dying and mourning rites seem incompatible. As a result, the rite she lived, which was supposed to satisfy her need for warmth, beauty, and humanity in a context marked by ugliness and impersonality, appear paradoxical. Her need to navigate between these irreconcilable fields gave rise to experiences of absurdity, as evoked at the end of her extract. Tania, for her part, seemed to have internalized this managerial logic, as shown by the technical and mechanical lexical field (in bold) used when she recounted the organization of her father's funeral:

> With my brother, for two days, [we had to] almost **evaluate** the relationship of friendship, family, to decide who deserved to be in the 15 people. (...) It became a bit of a **scientific calculation**, I don't know at all what **methodology** we defined, and when I had the final **list**, I had to call each person one by one. (...) And there I had no tact, because I had to call 15 people, so it was 'Thank you, you're one of the selected ones, you're coming or you're not, if you're coming, you'd better come. See you on Friday. Ok, next person.

Later, she added how '*horrible*' it was to '*be forced to turn into an automatic mode to handle everything*', revealing the emotional dissonance this forced state of mind had caused.

### How could we make sense of a fragmented rite?

Secondly, the rules forced people to modify the form of the rites by accelerating them, shortening them or fragmenting them, clashing with the whole process sought by the participants. Tania shared that restrictions had forced her to prepare

and perform rites quickly, as she had "*30 minutes on the clock*" for her father's funeral. Yet Louis explained how the time of the ceremony was necessary for the introspection that this kind of moment allows:

> And it's in those moments, in those rites (...), when you can take a break for, I don't know, an hour or two, (...), that's the moment when you can think about it and say to yourself, if... What wouldn't I have done? What would I have liked to do? What must I definitely do first? Should I tell someone? (…) These are the moments when it's all about that (…). When you imagine a death, you know that there's a subject-verb-completion. (...) And there, very clearly, there wasn't.

For him, rites need time to allow him to reflect on life. However, the rites he lived at that time lacked certain steps, preventing them from fulfilling their existential functions. Philippe, too, described his grandmother's funeral as incomplete:

> In a rite, things are already 'pre-set'. For example, I don't know, in a religious ceremony, there are stages, there's someone who speaks, then a reading, then this, that, then we say goodbye, and then we decide on the burial and so on. (...) But when there's none of that, you already arrive at the place where... I don't even know where she was before (...). I arrived alone and there was my grandmother, in a coffin, the funeral services, and a hole.

His extract shows how this truncated rite prevented him from making sense of his grandmother's death, leading to feelings of uncertainty and creating an incomplete narrative. As a result, like Henri, he experienced her death as a '*disappearance*':

> It's more like a disappearance when in fact she died under very normal conditions for a person of her age in a nursing home following an illness. She was 100. It was normal, it was expected, it wasn't anything. It's not like a mother disappearing or someone taking to the streets and that's it. But I think that in the way it happened, it was more like a disappearance than a death.

By insisting on how his grandmother's death was natural and expected for a long time, Philippe revealed how illogical this sensation of disappearance was for him, and the nonsense this created. Louis illustrated this unreality and this confusion by drawing an analogy with his memory of the birth of his children by caesarean section:

> This feeling of unreality (...), of not having been able to do what everyone else did, to take care of the body, to make the preparations. All that, not having lived it... I have an analogy with that: my children (...), it must have been a caesarean every time. And so, the 'um... what? It is now? But I haven't experienced it!' Well, it's more or less the same.

In that excerpt, the difficulty of making sense of his experience seems to stem from how quickly it happened, but also from the fact that it had not experienced what he expected to live at that time. There were no points of reference to which he could associate his experience and integrate it. Maria, for her part, explained that the virtual rite after her friend's death made her feel like the death was '*not tangible*'. To help her concretize her friend's death, she had kept the death notice on her fridge, something she had never done before:

> Every time, I'd say to myself, this is an obituary notice, that's why it's there. So, I think it was important for me to keep it because every time, and not in the creepy sense of the thing, but it reminded me of reality. And I think I probably needed it, given the particular context.

In conclusion, different participants questioned the meaning that these modified rites could still have for them. For Philippe, this fragmented rite '*doesn't allow you to do the work you have to do or to live. In the end you feel like you haven't really gone through anything. We didn't experience anything*'. Interestingly, this expression of *not having experienced anything* was also used by Louis in the extract above. This shows how these unsatisfactory modified rites, instead

of helping them to understand, left even more holes and confusion in their stories, leading to a sense of unreality and meaninglessness.

### Everything was put on hold, although it was now or never

During stricter periods, social and health restrictions suspended any possibility of collective rites, forcing people to find alternatives or to postpone them. However, this 'postponement' contradicted the participants' experience of a unique window of time, specific to the few days surrounding the death, which does not reappear later on and in which farewell rites and gatherings are found necessary. As Philippe said: '*when someone dies, you can't do it again. It's not like when you're going to redo your kitchen, then you can wait six months*'. This period of time around the death appeared as unique because of the psychological state they were in. Jeanne, for example, recounted the week following her father's death, insisting on how it was '*very special, as if time had stopped*', as if '*life and death [were] crossing paths, intertwining*' and recalling herself as being '*only in the present, present, present, impossible to think of my life before, impossible to think of even a week later*'.. In Jeanne and Louis's accounts, the specificity of this period also lay in the presence of the deceased's body, which contributed to the special atmosphere and without which there was something missing in the rite, as Louis explained: '*there was something about the rites, (…) that wasn't really there because we weren't able to form one body around my mum, we weren't able to do that.*'

### It made no sense to carry out social rites without getting together

All the participants stressed their need to be surrounded, to share rites with others and to have comforting physical contact, whereas the restrictions reduced the number of people allowed and prevented physical contact and face-to-face meetings. Louis expressed quite clearly that the injunction of performing rites without being together was a paradox for him, pointing out the nonsense and absurdity of rites without social contact: '*We're talking about ancestral rites, aren't we? The aim of the game is social cohesion, isn't it? But then, what's the point of burying someone with religious rites and all that if we can't get together?'.* For Jeanne, the need for social contact was instinctive and vital, making it even more incompatible with these distancing rules: '*I didn't have much of a sense of "me" and I couldn't really be on my own (…). I couldn't sleep on my own or even go for a walk on my own (...). It was very instinctive behaviors like touching each other, sleeping next to each other*'.

In conclusion, the rules imposed, by appearing contradictory or even totally incompatible with instinctive needs or representations of mourning rites, acted as paradoxical injunctions: "find humanity in technicalized and sanitized rites", "honor your dead but quickly", "perform social rites but without getting together". Consequently, this led to experiences of paradoxical rites, combining irreconcilable demands and needs and giving rise to feelings of absurdity, incomprehension, emotional dissonance or loss of meaning.

### Experiences were very different from what had been imagined, for better or for worse

Eleven of the twelve participants' accounts revealed an experience of a gap between what was expected and what was possible around the death of their loved one, as Dominique summed up: '*So really the influence of Covid was the difficulty there was between what we could do at the time, with the restrictions, and what we would have liked to do*'. However, the extent, nature and significance of this gap was specific to each person, as what was expected and what was possible differed from person to person. The three sub-themes reveal the diversity of experiences in terms of expectations, possible/impossible actions and meanings.

### A multitude of different expectations: between the ideal goodbye, the collective imagination and prescribed norms

The meaning people gave to their experience depended in part on what they had initially hoped for and what it meant to them. In their accounts, several people referred to what they had imagined they would do for their loved one. Not only did

the content vary, but so did what seemed to drive this imagination and the value or affect attached to it. For some, what was expected came from the explicit wishes of the deceased or their own wishes, driven by a hope or an ideal. Isabelle hoped to donate her father's body to science, as he had wished. Tania, for her part, had long imagined how she would like her father's funeral to be: '*It's something that's been going through our heads all these years, saying: "Well, the day Dad dies, what am I going to do? Who am I going to invite?" I imagined, I don't know, a funeral for a king*.' For others, what was expected was fed by collective visions of what a rite should look like, with no specific desire to follow these standards. Without the restrictions, for example, Élise would have '*done it the old-fashioned way. Meet at the church at 10. And then a sandwich in the hall*', and André would have had '*an ordinary funeral, [as] you sometimes see in the movies, the hearse, with 2 people following, the rain, the whole thing*'. Louis, too, had imagined '*that moment of shock, the wake, when you welcome people who come to visit. I'm putting coffee back in the thermos, welcoming people... I didn't have that.*' In parallel, the importance of rites following prescribed norms varied from story to story. While for Élise and André, the restrictions offered an opportunity to '*leave room for whatever comes*', for others, it was important for their way of saying goodbye to be rooted in tradition. When Jeanne and her family asked themselves how to create a rite, they concluded:

> In terms of rites, my brothers said, "we're not going to do it in church", so we said, "we'll do it in a field". And in fact, all of a sudden we said to ourselves, "you can't just make that up, how do you do a ceremony, who's going to do what?", and then we said, "yes, rites are there for a reason, we are supposed to follow them, because you can't just invent something like that".

As this extract shows, the circumstances that limited or changed rites raised questions that needed to be answered: *How do we construct a rite? Why and for whom are we doing it? What counts in the rite? Who has a place in this rite?* The answers to these questions varied from person to person, revealing what was considered important in those moments. As a result, what was prevented by restrictions took on a singular meaning. For example, Tania found it particularly difficult not to be allowed to hold a ceremony with a large audience, as this would have been a way of honoring her father's life and offering him something in return for what he had done for his children. Maria and Philippe regretted the impossibility of meeting and touching their relatives, while Louis deplored the lack of an informal moment, all three insisting on the importance of the rite to share memories, to support and care for each other. Isabelle and her siblings, for their part, had not planned a funeral after their father's euthanasia, as they did not consider them meaningful. Funeral restrictions were therefore of little importance to them.

### A multitude of different possibilities for action: from freedom to powerlessness

Faced with the restrictions on possible actions and gestures, what was possible to do took on different meanings and realities for each participant, leading to various feelings of control and agency. Experience of one's agency seemed to be the result of a balance between the subjective importance and legitimacy accorded to restrictions and their perception of concrete and practical conditions to which they were subjected.

Firstly, our participants had very different subjective perceptions of the legitimacy and the importance of the restrictions, mostly regarding moral (are the restrictions acceptable and moral?) and legal aspects (is compliance with the law important?). For example, Jeanne, Élise and Philippe pointed out the immorality and the violence of the restrictions. Others, like Maria and Dominique, found them acceptable and justified, although, as Maria added: '*Just because we don't find them too strict doesn't mean we're satisfied with them. You can accept them and still feel that the situation is not ideal*'. In legal terms, the relationship with the law colored the subjective experience of complying (or not) with the restrictions. For example, Tania felt '*like a lawbreaker*' and '*extremely guilty*' for holding her father's hand when it was not allowed. The subjective positioning towards restrictions formed a representation that could be ambivalent and complex. Moreover, although most of the participants considered the restrictions and reduction of contamination important, some recounted how these restrictions lost their significance in the context of mourning. In the stories told by some participants, the death

of their loved one shifted the order of priorities, as Élise when her 8-year-old son died: '*Everything I believed in, every-thing I thought was important, fell apart, including the Covid rules*'. Jeanne also described how non-compliance appeared obvious despite the importance she and her family attached to the measures before her father's death: '*There are a lot of doctors in my family, so they're really people who follow instructions (laughs). And there, there wasn't even a question of... Even my great aunt, there was no question of talking to her about distancing*'. Tania, for her part, noted how surprising and paradoxical it was for her when the restrictions suddenly lost their importance:

> It's really paradoxical because I was the first, from the very first wave, to understand how dangerous it was, how viral, how we had to be careful, but there's nothing you can do, when something like that happens to you, you don't give a damn about all these rules.

Secondly, the experiences of our participants were also shaped by the external conditions and their perceptions of them, which fueled their sense of control over the situation. For three participants, the police or funeral directors allowed exceptions regarding the length of the ceremony or the number of people allowed to attend. In contrast, Naomi shared an account tinged with resentment, in which the police had come to check the number of people present on the day of the funeral. Tania, too, felt a sense of injustice in the face of the many refusals from those involved: '*not once did I have the impression that a single doctor or nurse had a bit of pity, a bit of empathy, I really didn't get anything at all*'.

Thirdly, in addition to their perceptions of restrictions and external conditions, some participants described feeling stupefied and having difficulty taking action. Philippe, for example, explained how difficult it was for him and his family to organize the funeral, and how it led to moments of disorganization, creating "*a kind of completely absurd thing*". Later, he said that he was unable to react when the nursing home burned his grandmother's belongings:

> I didn't have the courage to do anything and at the same time I thought, what's the point? At least, if it had been today, it wouldn't have been any use, but she would have heard me already and I think it would have helped me to just go and say 'but what is this? It's nonsense, it doesn't make any sense'.

In conclusion, fueled by their subjective experiences and the external conditions, the possibility for some people, like Sandra, to take '*all the freedom we could*', contrasted with the experiences of Philippe, Naomi and Tania, who felt powerless or stupefied in the face of the measures, and had the impression that '*we had no choice but to comply'* as Naomi put it.

### A multitude of meanings: from the extraordinary to the stolen goodbye

The narrative surrounding the death and the emotions and meanings associated with it varied greatly, from an exceptional to a stolen farewell. Several people said that they had experienced an extraordinary goodbye, more intimate, authentic or spontaneous, which would probably not have happened in any other context. For Élise, who lost her son, the circumstances allowed her to '*let go and make room for whatever came'*, and to live a meaningful moment:

> And in fact, it was the Covid rules that made this possible. We would never have taken this great approach (...). We would have done it the old-fashioned way. Come up with something stupid that wouldn't have made sense. But here everything made sense. And it was so much like him, at home, in his garden. This is where he was conceived. This is where he was born, this is where he died. I mean, it's crazy.

André also noted that this specific context allowed for an intimate, creative and powerful goodbye, that helped him in his grieving process:

> I think if it had been a normal funeral I wouldn't have experienced it like this (...) It was a final farewell with the family, it was powerful. I came out of it, I won't say the mourning was over, but I was relieved. The goodbye was over. I came out of it [he insists] serene.

On the contrary, other participants shared the feeling that they had been prevented from saying goodbye properly. When what was done did not meet their expectations, when the alternatives seemed insufficient or meaningless, when restrictions were perceived as unacceptable, and when they felt limited or powerless, participants seemed less able to find solace in the rites. Philippe found this farewell disrespectful for his grandma and Tania felt '*so disappointed and defeated by this farewell*', adding that her father '*deserved better*'. However, not everyone said they were affected. Louis nuanced his comments by saying '*I didn't live through all that (...) Do I miss it? No. It's a story, that's all*', which echoes Henri's words: '*We put it in a plastic bag and left. And that, even though I'm not a family man, shocks me terribly. But sad? I don't know*'.

In addition to the impact on the goodbye, restrictive circumstances also affected people's grief in other ways. For some, they have influenced the occurrence of the death itself, by worsening health conditions or limiting care or social support, like for Sandra's daughter and Jeanne's father. For others, they had an impact on longer-term grieving processes. In this respect, Tania and Naomi, who had a very negative experience of their father's farewell rites because of the restrictions, explained how much these same measures had helped them in their grief, allowing them respectively '*not to have to pretend*' and '*to feel more contained*'. Élise benefited from the greater availability of those close to her, and André said he appreciated being able to grieve '*without any troublemakers around*'. Jeanne, on the other hand, felt more vulnerable and weakened because the restrictions isolated her and reduced her activities. In conclusion, between different people and even within the same story, the experience was multiple and complex, difficult to reduce to a positive or negative impact of the circumstances or to satisfaction or dissatisfaction with the goodbye.

### Experiences had multiple and evolving meanings

These experiences, very different from what had been imagined and sometimes unreal and incomplete, had to be given meaning. Because of the diversity of experiences and the novelty of the context, the search for meaning seems to have been a personal quest, with no common points of reference available. Different trajectories of narratives emerged; some moving towards appeasement, others marked by constant anger, and are represented in the first two themes. In addition, these trajectories combined various levels of meaning: relational, moral, existential, that are represented in the two last subthemes. The subthemes are illustrated by a participant's extract, but as explained for each, their meanings can be more broadly applied to other participants' experiences. This theme incorporates a temporal aspect, moving away from the moment of experience to capture how people made sense of what they had experienced and what, in their narratives, changed or remained the same over time.

### "I am at peace now"

Various participants showed how their experience and interpretation of what they had lived through had changed over time towards a sense of appeasement. Élise was '*at peace*', considering that '*we did what we could*', while Sandra summed up: '*It's as if it was normal, as if that's how it should be*'. Over time, they found ways to give a coherent and acceptable sense of their experiences. Firstly, when what had been done at the time of death was unsatisfactory, incomplete or unacceptable, some participants took concrete action to compensate. Élise, who felt that the farewell to her son had not been sufficient, was later able to devise and organize a ceremony that was appropriate for him, thus making up for her regrets: '*And now I'm at peace with myself. I've done... That's what I was saying about any regrets I may have had, well... (...) Celebrating afterwards allowed me to do... what I wanted to do*'.. Secondly, several people sought to situate their experience in relation to others. In this unprecedented context, other people's experiences seemed to provide reference points for assessing the normality/abnormality or seriousness of their own experience. Therefore, some participants put things into perspective by comparing it to situations they considered worse, experienced by themselves or others. Dominique relativized his wife's funeral by thinking back to his aunt's funeral, where the constraints were more severe. He insisted: '*It's not like the people who had Covid and died without being able to see their family again. Yes, it was not like that, ok?*'. Isabelle also compared her experience with that of other people:

Some people are still very sad because they weren't able to say goodbye the way they wanted to, they felt like they had finally been taken away from the last moments of their loved ones. And I say to myself, that, that must be terrible. (...) In the end, we're happy with the way things turned out.

The last sentence of this quotation reveals that comparing her experience with others helped Isabelle to reinterpret what she had experienced and to orient her emotional experience, concluding that, in the end, she could be happy with it. André, too, emphasized how '*lucky and honored*' he was to have been invited to the ceremony when others could not be there, and how grateful for that he was. Thirdly, some participants reinterpreted their experience by choosing, more or less consciously, the memories, feelings, meanings and possible explanations put forward in their accounts, namely the responsibility of the covid situation (underlined in the next excerpts). Élise, when many of her relatives attributed her son's death – by accident – to the circumstances of lockdown, explained that she preferred to consciously decide on a narrative:

So, what I like to say to myself, and that helps me with what happened, with the guilt, with the grief I'm going through (...). I prefer to tell myself that it was [supposed to be] like that rather than tell myself that it could have been avoided, because that's worse. Let's say he had to die that day (...), well 'Thank you Covid' because he died at home, in a place he built with his dad, in his mum's arms.

Some people also described how elements at one point were significant and then had lost importance or had been interpreted differently. Élise, for example, said that certain obstacles, such as not being able to print a photo of her son for the ceremony, were '*a catastrophe*', but '*afterwards, it's not a big deal*'. Jeanne, talking about the ceremony, added '*afterwards, we said to ourselves that it would not have been so intimate, (…) that maybe, it wasn't so bad after all*'. André, in turn, noted a change in his interpretation of his romantic partner's death by suicide. '*At first*', he explained, '*I was complaining about this bloody Covid that prevented me from holding her hand, [thinking] that she might still be there*', but that over time, he had changed his interpretation, considering that she would not necessarily have been better in the end, and that it was ultimately her choice.

In conclusion, some participants like Élise, Sandra, André, Isabelle and Dominique were able to make sense of the death of their loved one in restrictive circumstances by repairing, relativizing, or reinterpreting their experience in a way that brought them peace or gratitude. This subtheme, by revealing the evolving and dynamic nature of meaning, also shows that these narratives may change in the future, allowing other emotions and significations to emerge.

**"I'll be angry all my life"**

In contrast, other participants expressed unrelenting feelings of anger, injustice, and unacceptability, revealing a difficulty in making sense of what they had experienced. Tania and Naomi shared feelings of anger that had not left them by the time of the interview. As Tania put it: '*I have this knot in my stomach that makes it difficult to move on and feel less angry*', reflecting how this constant anger immobilized her and prevented her from moving forward. She later explained that she had considered organizing a postponed rite, thinking that it might help her soothing her anger, but she was afraid she could not bear another disappointment:

I regret so much not having experienced that moment at the right time, that when I think about it, I say to myself: "Well, you're still very angry, it would do you good to make up for it by having a ceremony where you can invite everyone". In fact, what I fear most is (...) that it again won't happen in the way I'd imagined. I've got the impression that it could be a double-edged sword: either it goes really well and some of my anger eases and I feel really good about it, or it's going to make me even angrier.

This excerpt reflects her sense of powerlessness over her own feelings and what she could do to soothe them, echoing her lack of control over her father's illness, death, and funerals two years before and mentioned in the previous theme. This contrasts also the sense of control that Elise and André seemed to have on the creation of their own narrative, when they chose, in the precedent subtheme, what to put forward in their accounts. Philippe, for his part, was also '*upset for a long time afterwards*', even if his anger had diminished since then because '*time erases things*' and because he had been trying not to think about that difficult period overall. He added '*if we were doing well at that time, it would be okay to remember*'. This difficulty in thinking about the lived experience was also shared by Tania and Naomi, who respectively explained that they had resorted to antidepressants and alcohol to cope with the overwhelming feelings following their fathers' death. Yet, Tania pointed out how it was not supposed to be that difficult as she had been expecting her father's death, and had been preparing for it for years, revealing how difficult it is for her to make sense of that *illogical* experience:

> It's horrible, but it's true that, psychologically, I was ready for my father not to live for years. I was already psychologically prepared for what I was going to experience in a few years' time, but nothing went as planned. Everything went horribly, inhumanly and illogically, in fact. And that's what makes me so angry and I feel a sense of injustice.

For Tania and Naomi, that anger came with the issue of responsibility. Naomi consciously chose to blame the hospital in order to direct her anger toward someone other than her own family:

> I blame life, I blame the whole world. There is always someone to blame. I blame the hospital. I know they did everything (...) but it's my own way of reacting and choosing to blame them so that I don't blame the whole world and even less my loved ones who are in the same pain as me. So, it helps me to put the blame somewhere.

For Tania, the responsibility for her father's limited funerals was a complex issue she was still trying to figure out: '*I'm angry, but I don't know who I'm angry with, but I'll be angry all my life. It's like the government, the people who made those decisions, for not allowing me to bury my father with dignity*'. Yet, her account was also tinged with a sense of guilt, when she said: '*the only thing I could offer him was to bury him in the same grave as our mother, but fortunately that was the only thing I was able to do properly*', although adding '*even though I know it's not my fault*'. ' In contrast to the experiences of Dominique and Isabelle, who put things into perspective by comparing their experiences with those of people who had gone through "worse", Tania, recounted the feelings of injustice and jealousy she felt when comparing her experience to others, and her resulting need for recognition:

> In the months after my father's death, I was sometimes jealous of the people who were burying their loved ones when they started to increase the number of people allowed to attend. I was like, 'Oh, those people, they can invite 50 people, they're so lucky. (...) I mean, it's an absurd thing to say, but I really thought that. (...) Obviously, we talk about those who were victims of Covid. My father didn't die of Covid, but I have the impression that no one has ever taken an interest in all the people who have been affected by this situation

This subtheme shows how some participants felt overwhelmed and paralyzed by their anger, struggling to make sense of their experiences. It also reveals how similar processes of meaning-making may influence narratives and emotions in very different ways from person to person: they can be soothing for some, while fueling anger, fear and injustice in others.

**"This is death for society? It's horrible, horrible, horrible"**

Until now, the subthemes illustrated the meanings participants gave to their own personal experience and the one of their relatives: what they expected, what they were able to do or not do, how they were able to meet their needs and their loved ones and how it affected them. Other participants also incorporated a moral reading of the situation they lived,

which colored their emotional experience. Jeanne for example, recounted on several occasions how deeply she had been affected by the treatment of death and dying she had observed. Over and above her personal experience of her father's death and what she and those close to her had been able to do, which overall were satisfying, what it said about the society strongly affected her and aroused a different kind of distress:

> I think I have some... I don't know how to say it but a moral, a strong sense where I say to myself *'these are things that shouldn't be like this'*, I feel it strongly so there I was too, I said to myself '*this is death for society, it's horrible, it's horrible, it's horrible'*.

Philippe, too, shared his indignation: he felt that the measures imposed by the authorities on funeral rites '*touched on our humanity'*, were '*almost anticonstitutional'* and revealed a great '*lack of respect for the dead'* and for traditions.

### "I don't prefer my life now, but I prefer my vision of life"

Another way of reading their own experience was by identifying what it had taught them about what matters when a relative is dying or in life in general. This meaning, more broadly existential, was brought by several participants. Some of them explained that their experience had changed what they thought was important on the death of a loved one. Philippe, for example, explained: *'I realized that the rite was necessary. I knew that before, but the fact that I wasn't allowed to carry it out. (...) I experienced it for the first time'*. André, who experienced a very intimate farewell, said that it made him question the norms surrounding rites even more: '*Why do we have to make a funeral face at funerals? If we don't have to follow a pre-established rite that might not suit us and we can do what we really want to do, I think that's good'*. Louis, for his part, '*realized that we could do without this solemn thing'* and confirmed that '*that's what the rite is about, the important thing is the family reunion'*. Elise and Jeanne, for their part, explained that it had changed them and their vision of life. Jeanne explained how it changed her relationship to death, creating an openness to talk about it and a closeness with it, as well as a new sensitivity to the world. Élise described how it changed her vision or life, her consideration of what really matters and what does not: '*I don't prefer my life now, but I prefer my vision of life now'*. Both of their experiences revealed a sense of existential growth, adding a new dimension to the meanings they already shared.

These subthemes show that the meaning the participants gave to their experience were rooted in their personal experience as well as in a more moral and existential reading of what it says about society, life, death and existence. The emotional experience that emerges therefore appears complex, as it intertwines emotions that may be different on distinct levels of meanings.

## Discussion

With the aim of understanding the meanings that people gave to their experience of the death of a loved one and of grieving in circumstances where funeral rites and social gatherings were restricted, we carried out an interpretative phenomenological analysis of 12 interviews with bereaved people. What emerged was that experiencing the death of their loved one in these circumstances meant that (a) contradictions between pandemic and mourning fields acted as paradoxical injunctions, (b) experiences were very different from what had been imagined, for better or for worse, and (c) experiences had multiple and evolving meanings. Overall, some of our findings align with the literature on grief in this specific context, in particular the impression of a disappearance rather than a death, of unreality and intangibility of the death due to the distance and lack of rites [64], the feelings of an uncomplete and fragmented experience [65], and, for some, a grieving experience marked by a lack of social acknowledgement [25,66]. That said, our results also provide new elements on which our discussion will focus.

The first theme of our findings highlights the contradictions between, on the one hand, the managerial and dehumanizing, accelerated and dissocialized approach to rites and dying imposed by the restrictions and, on the other hand, our

participants's needs for humanity, beauty, sacredness, time, wholeness, and social and emotional closeness. Although limited to the context of the pandemic, they recall the antinomy Des Aulniers [67] noted between the rhythm and depth required by funeral rites and the speed and technicality that sometimes characterize contemporary rites in recent decades. However, the difference here is that, in the participants' experiences, the technical and impersonal aspects were imposed by the restrictions and did not result from their own initiative in a process of progressive evolution. These aspects were in conflict with needs or obligations, whether external or internal, such as internalized norms, ethical or moral principles, or representations of *how rites should be*. As a result, having to live meaningful mourning rites in this context was experienced by some as paradoxical injunctions: *find humanity in technicalized and sanitized rites, find meaning in fragmented rites, honor your dead but quickly, perform social rites but without getting together*. Paradoxical injunctions, a concept developed by the anthropologist Bateson and taken up in systemic psychotherapy and in clinical sociology [68–69], are defined as the imposition of a double constraint, made up of obligatory and mutually exclusive elements, and therefore impossible to satisfy. This contradiction, sometimes invisible, can create feelings of anguish, fear, guilt or paralysis, as the person is "*caught between two impossible demands*" [68, p. 366]. This is consistent with our findings showing feelings of guilt, stupefaction and difficulty in taking action. Paradoxical injunctions can also produce internal conflict and emotional dissonance, as shown in Tania's experience, when the injunctions were internalized. For people who had to combine these incompatible demands and needs, rites were experienced as paradoxical, absurd, illogical, and gave rise to feelings of incomprehension, disappointment, and loss of meaning. This raises the question of what function rites marked by so many contradictions can still fulfil. Other participants, however, were able to find solace in these constrained rites, noting the intimacy and authenticity they enabled. In that sense, our findings align with Burrell and Selman's [11] reflection on the importance of the meaning rather than the form of rites. Our results suggest that this meaning is more likely to emerge when people perceived (or regained) a certain degree of freedom and agency. Conversely, powerlessness and the feeling of not having had a choice seem to contribute to the impression of '*having experienced nothing*'.

The second theme of the study reveals that the experiences our participants had of the death of their loved and their mourning rites were far from what they had originally expected. The extent, nature and significance of this gap was specific to each person, as what was expected and what was possible differed from person to person. In terms of expectations, while some people had a representation of the ideal goodbye, others based their expectations on collective notions or on internalized norms or traditions, with various affects associated. As these expectations had a specific and idiosyncratic meaning for each person, their restrictions had different consequences and meanings, as Long et al. [20] have also shown. In terms of possibilities, even during a similar period of restrictions, what could be done appeared to be specific to each person. Their sense of agency or powerlessness were influenced by their subjective representations of the restrictions, the external actors regulating their application, and a possible feeling of stupefaction as mentioned above. Consequently, some people felt that what they had done was better than they had imagined, while, for others, the obstacles were too great, and the discrepancy gave way to regrets, a sense of a missed or failed goodbye. This echoes the experience of '*violation of grief*' highlighted by Cipolletta et al. [22], when people were unable to carry out the rites they wanted for their loved one. In our results, this feeling seems to be reinforced by the idea that '*you only die once*' and that '*you can't do it again*', and by the impression that the days surrounding the death were a unique and specific moment in which rites and social and emotional gestures were indispensable. This resonates with the Greek notion of *kairos*, which describes the ideal moment before which it is too soon and after which it is too late. In line with these ideas, we hypothesize that for some people, the fact that they were not able to carry out the desired gestures or actions around the death of their loved one led to a form of '*grieving for the ideal farewell*', especially when they considered that these actions should absolutely have taken place in the days surrounding the death and that, after that, it would have been too late. This echoes a recent study that showed that the loss of a relative during the pandemic was compounded by another loss: that of the desired farewell, resulting in overshadowed grief [70]. This grief for the ideal goodbye may be a particular form of *unfinished business* [71]. Our findings suggest that the impression of having missed out on this goodbye may lead to

feelings of guilt towards the deceased or towards other bereaved people, but may also play out on a psychological, moral or existential level. This hypothesis could help us understand the feelings of self-blame that many bereaved people experienced during the pandemic [72] and that are strongly linked to situations of unfinished business [73].

The first two themes, highlighting paradoxical rites and unfulfilled expectations, showed that people's representations had been shattered. In the literature on meaning-making, these *shattered assumptions* [46] arise when the experience does not match the set of beliefs, expectations, values, and representations with which the individual makes sense of the world. They are the conditions for triggering processes of meaning reconstruction [45], whose aim is to reduce this incongruence [47]. The third theme highlighted that the participants needed to make sense of what they had experienced. It also showed that those meanings were constructed in different ways for each person, evolved over time and were made up of several levels of meaning. First, the participants attempted to make sense of their experiences through various processes. While some people took action to remedy what could not have been done, others tried to make sense to it by comparing it to other situations, by emphasizing certain elements in their account rather than others, or by (re)attributing responsibility for the situation. Whatever the process, the meaning that was made varied from person to person and led to very different emotions, sometimes intertwined: relief, ambivalence, bitterness, anger, regret, gratitude, jealousy or a need for acknowledgement. This underlines the fact that the processes by which meaning is given are diverse, and that they do not result in the same meaning or the same feeling for everyone: what people remember, interpret, and the resulting emotions differ greatly depending to the meaning that they have or have not managed to give to their experience. These results are consistent with a study that demonstrated the mediating role of meaning-making processes between the circumstances of the pandemic and grief reactions [29]. Those processes were often constructed through interaction with others: by reacting to other people's discourse, comparing their experiences to others in similar contexts, and retelling their own story. This demonstrates how meanings are not merely individual, intrapsychic processes, isolated from the social context in which they are created. Instead, they are co-created through interactions and shared narratives shaped by specific historical and social contexts, as Neimeyer, Klass and Dennis argued in their Social Constructionist Account of Grief [74]. In addition, our findings supported the evolving and dynamic nature of meaning, as some participants highlighted how their interpretation of their experiences had changed over time. Each participant had their own '*meaning trajectory'*, an account of their experiences that evolved through different processes, as well as the emotional experience of it. Thirdly, our findings showed that our participants' narratives were composed of different levels of meanings. While most participants recounted their experience by making sense of their personal experience, focusing on their situation, their grief and their relationship with the deceased, others added a more moral or existential reading to their experience, by analyzing what their experiences revealed about the society, about what matters when a loved one dies, and about the existence in a broader sense. These shattered existential assumptions sparked growth or indignation and were interwoven with other levels of meaning to form a complex emotional experience. This illustrates the multitude of meanings assigned to an experience, aligning with Bonanno & Kaltman's conceptualization of a *continuum of subjective meanings* [35].

In conclusion, faced with the paradoxical injunctions of the pandemic context, each bereaved participant had a singular experience, far from what they had expected. While some experienced paradoxical rites to which it was difficult to give meaning, others found solace through different processes. The diversity of experiences of the death of a relative in restrictive circumstances has already been noted in other qualitative studies [19,20]. However, the present study sheds additional light on this issue by suggesting various aspects that make these experiences unique, namely the initial appraisals, expectations and meanings given to the rites, as well as to the restrictions, or the perceived possibilities for action. As such, our findings align with Long and colleagues' suggestion to approach funeral restrictions in a neutral way, seeing them as a "*disruption*" rather than a deprivation [20]. This approach aims to remain in an exploratory stance, allowing us to capture people's experiences in all their diversity, whether they experienced these restrictions with distress, relief, strangeness, ambivalence or as an opportunity to reinvent and reappropriate practices. Moreover, our findings show the diversity

that exists within a single account, revealing each person's experience as complex and ambivalent. On the one hand, as the experience of bereavement is multiple and multidimensional, the context may affect certain aspects or certain periods differently [75]: some participants felt that the circumstances had made the death of their loved one *'unfair'* and *'horrible'*, while at the same time feeling that they had helped them in their subsequent grief, or vice versa. On the other hand, some people explained that the situation had challenged their view of the world and society, provoking feelings of injustice or anger at a more existential level, without this being expressed in relation to their bereavement. In short, not only can circumstances affect the bereavement experience at different levels, but they can also affect people's experiences, meanings and representations without necessarily influencing the person's bereavement or their bond with the deceased.

## Clinical and research implications

Our findings open up clinical perspectives. First, our findings showed that the contradictions between the fields of the pandemic and mourning might have acted as paradoxical injunctions. This provides avenues for understanding and addressing the feelings of guilt, absurdity, stupefaction, and powerlessness that were observed in some participants and that can emerge when these paradoxical injunctions are not perceived or acknowledged [68,69]. In a therapeutic setting, putting light on the incompatibility of the elements they contained, recognizing the impossibility of reconciling both may help to identify new points of reference that restore meaning and relieve the person of guilt. Secondly, our findings present the notion of grieving the ideal goodbye as a new perspective for understanding the grieving processes and possible forms of unfinished business that bereaved people may experience. They encourage clinicians to explore the gap between what was hoped for and what was done, and the meaning of renouncing this ideal goodbye, as well as to help the person disentangle feelings related to their loved one's death from those related to this renouncement. Our results showed that feelings of powerlessness might play a key role in the difficulty of grieving the ideal goodbye. Participants who felt powerless and felt that they did not have any choice recounted a farewell that was unacceptable for them, marked by a profound sense of injustice and unrelenting anger. This trapped them in an unevolving narrative, making it difficult to make sense of their experience or take potentially soothing actions. In that sense, therapeutic work may help them find agency and closure by providing a secure setting in which to experience potentially healing moments that make sense to them, such as individual or collective recreated rites, elegy or memorialization (see [76], part XVI). In general, our results show how idiosyncratic, complex, evolutive and diverse the experiences of the loss of a person and grief are, and therefore, encourage to focus on the experience of the person, without judgement or any attempt at prediction, recognizing and allowing contradictory options and ambivalent feelings to be experienced. In that sense, they support a person-centered approach [77–79] that acknowledges the singularity, subjectivity, and evolving nature of grief experiences and their sense within the person's frame of reference. They also align with Guldin and Leget's perspective [2], that insists on the multidimensionality of grief and its existential dimensions.

Our findings also have implications for bereavement research. First, they show how illuminating the notion of paradoxical injunctions is for understanding bereavement under restrictive conditions. This could be further examined in relation to the ambivalent and meaningless feelings that may be experienced in other bereavement circumstances where bereaved people may be caught into paradoxical injunctions, due to the opposition of cultural norms, familial expectations and demands, and moral principles. More broadly, our findings suggest the necessity of exploring the shattered assumptions and existential tensions that arise when facing the death of a loved one. This includes the search for meaning and purpose for one's life, which was already investigated [45], but also less obvious existential reflections about loneliness-belongingness, freedom-ineluctability of one and close one's life and partial controllability over what happens, with representations of moral concerns and socially constructed norms about what is right and wrong when someone dies [80]. Then, they encourage further exploration of grieving for the ideal goodbye as a form of unfinished business, and its potential relationship to people's conception of a "good death" [81]. Our findings highlight that the meanings our participants gave to their experiences were strongly colored by their perception of agency and controllability they had, as

well as the fairness and morality of what they have lived. They also show that these perceptions might evolve over time, as well as the meanings and emotional experiences, and that they are anchored in historical, social and cultural contexts and co-constructed within shared narratives and social interactions. This encourages researchers to address the dynamic and subjective nature of grief processes, by including mediators in their studies, such as appraisals of the circumstances of death in terms of uncontrollability, injustice, general valence, as well as other mediators of meanings. The diversity and complexity of the experiences shown in our findings make it difficult to summarize them in terms of negative or positive impacts of the circumstances, or in terms of an increase or decrease in the presence of 'symptoms'. In this way, they offer a possible explanation for the contradictions that appeared in the studies assessing the impact of the circumstances of covid on the grief experiences. This great diversity calls for caution when generalizing shared experiences, and even more so when predicting a general common effect. In this sense, our results encourage the adoption of methods that focus on the person's singular experience, such as phenomenological qualitative approaches or person-specific approaches [82], or at least that recognize the heterogeneity between experiences, such as person-centered quantitative research methods, like latent profile analyses or growth mixture models [82,83]. In addition, they lead us to broaden our understanding of the concept of *impact* and to question its study by bidirectional measures linking one or more predictors (e.g., level of restrictions, cause of death) and a dependent variable measuring an isolated indicator of grief (e.g., intensity of grief reactions, prevalence of PGD). As an example, among the participants who scored high on the TGI-SR, suggesting a high intensity of grief, some reported positive and meaningful experiences of the funeral and farewells, said they were '*at peace*' with the way things went after the death, while others gave an account tinged with anger and resentment. Similarly, among those who scored low on the TGI-SR, some shared their anger, had difficulty finding meaning and tried to avoid thinking about the death, while others were serene. These results call for caution when classifying reactions that appear to be similar on the basis of scales, and, even more so, when pathologizing them.

## Limitations

Despite its original contributions, our study has limitations that should be kept in mind. First, the interviewees were selected to obtain a diverse sample in terms of profile and experience. Logically, the diversity found in the results can be linked to this selection process. However, we observed that this diversity extended beyond these selection criteria, even among experiences that could be considered similar according to these criteria. Second, when the interview guide was created, the research question did not initially focus on the meaning given to the experiences. The theoretical framework was adopted after the interviews. Therefore, the guide did not explicitly include questions about meaning. That said, direct questioning is not necessarily required, as meanings are largely latent. One way of capturing them is to explore people's experiences from their own perspective and to access their narrative as openly and fully as possible [55], which was the aim of the interview guide. We recognize that our study contains a large sample for the IPA method. However, during the analysis, we took the necessary time to thoroughly and idiographically immerse ourselves in each interview, analyzing each extract individually to ensure our analysis reflected the lived experience of all the participants. Furthermore, although we encouraged a holistic perspective, our study sought to capture experiences in the context of the pandemic To this end, the analyses focused on the elements of the narrative that were specific to these circumstances (e.g., restricted rites). For the sake of scientific clarity, this approach may have prevented a more holistic understanding of our participants' grieving experiences and their evolution over time. Finally, it should be noted that the researchers did not receive any feedback on the results from the participants.

## Conclusion

In conclusion, this study showed that the contradictions between the fields of pandemic and mourning may have acted as paradoxical injunctions for some participants. For those who had to combine these two irreconcilable fields, this led to paradoxical and absurd rites, which were difficult to make sense of, especially when they felt they had no choice or freedom. Conversely,

others were able to experience meaningful rites and found solace in the authenticity, intimacy and creativity allowed by the circumstances. The study suggests that the notions of paradoxical injunctions, the gap between what was imagined and possible and the grief for the ideal goodbye that may result may be illuminating for understanding experiences of the bereaved under pandemic. It also draws therapeutic avenues to address feelings of guilt, loss of meaning, stupefaction and powerlessness shown by some bereaved. Our results confirm the relevance of meaning-making as a theoretical framework, as they show how much people's representations had been shaken by their experiences and how they were able to find ways of making sense of them, revealing distress when it was difficult to find meaning or when the meaning found was unacceptable. This theoretical framework also makes it possible to recognize that people's representations of the world and society may have been affected, leading to feelings of injustice or anger, without these being expressed in relation to their bereavement.

In general, these experiences were diverse and complex, making it difficult to describe the impact of the circumstances as simply positive or negative. The results should encourage those working with bereavement, whether in research or clinical practice, to recognize the singularity, subjectivity, and complexity of these experiences, whatever the circumstances. In addition, they suggest that a more holistic understanding of the concept of impact should be adopted, and call into question the relevance of studying it by targeting isolated variables, particularly where bereavement is concerned.

## Supporting information

**S1 Checklist. Inclusivity in global research questionnaire.**
(DOCX)

**S1 Positioning. Authors' subjective positioning.**
(DOCX)

## Acknowledgments

Firstly, the authors would like to express their gratitude to the twelve participants included in the study. They would like to thank M. Hustinx for her help in conducting the interviews, as well as E. Mersch and N-M. Lievens for their help in transcribing the interviews. They would also like to thank the members of the PC-Lab for their recommendations and support.

## Author contributions

**Conceptualization:** Camille Boever, Emmanuelle Zech.

**Data curation:** Camille Boever.

**Formal analysis:** Camille Boever, Laurence Arcand, Chantal Verdon.

**Funding acquisition:** Camille Boever.

**Investigation:** Camille Boever.

**Methodology:** Camille Boever, Laurence Arcand, Chantal Verdon.

**Project administration:** Camille Boever, Emmanuelle Zech.

**Resources:** Camille Boever, Emmanuelle Zech.

**Supervision:** Emmanuelle Zech, Chantal Verdon.

**Validation:** Emmanuelle Zech, Laurence Arcand, Chantal Verdon.

**Visualization:** Emmanuelle Zech.

**Writing – original draft:** Camille Boever.

**Writing – review & editing:** Camille Boever, Emmanuelle Zech, Laurence Arcand, Chantal Verdon.

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
