## [Decision Letter · Decision Letter 0]

1 Jul 2025

PONE-D-25-18637What does it mean to have experienced the death of a relative in a context of social and funeral restrictions? Lessons from the pandemic for bereavement research and clinical practice.PLOS ONE?

Dear Dr. Boever,

Thank you for submitting your manuscript to PLOS ONE. After careful consideration, we feel that it has merit but does not fully meet PLOS ONE’s publication criteria as it currently stands. Therefore, we invite you to submit a revised version of the manuscript that addresses the points raised during the review process.

The reviewers have completed their assessment of your manuscript, and while they found it engaging, they raised several significant concerns that need to be addressed. I concur with the reviewers, particularly regarding methodological issues. Specifically, your description of the analytic process appears more consistent with thematic analysis than with interpretative phenomenology (IP), and your Results section tends toward description rather than interpretation. Additionally, I recommend you carefully reconsider the conceptual clarity around your use of the term "paradox," as it is frequently employed to characterize participants' dilemmas or experiential tensions.

We look forward to receiving your revised manuscript.

Kind regards,

Michal Mahat-Shamir, Ph.D.

Academic Editor

PLOS ONE

Journal Requirements:

“CB was supported by a Grant "Fonds pour la Recherche en Sciences Humaines" (FRESH) from the "Fonds de la recherche Scientifique" (F.N.R.S.-F.R.S.) [number 1.F.010.22F.]

website : https://www.frs-fnrs.be/fr/financements/chercheur-doctorant#fresh

They did not play any role in the study design or any step of the research.” 

“This research was generously supported by a FRESH (F.N.R.S.-F.R.S.) Grant to CB [number 1.F.010.22F.]”

“CB was supported by a Grant "Fonds pour la Recherche en Sciences Humaines" (FRESH) from the "Fonds de la recherche Scientifique" (F.N.R.S.-F.R.S.) [number 1.F.010.22F.]

website : https://www.frs-fnrs.be/fr/financements/chercheur-doctorant#fresh

They did not play any role in the study design or any step of the research.”

6. Please remove all personal information, ensure that the data shared are in accordance with participant consent, and re-upload a fully anonymized data set.

Reviewers' comments:

Reviewer's Responses to Questions

**Comments to the Author**

1. Is the manuscript technically sound, and do the data support the conclusions?

Reviewer #1: Partly

Reviewer #2: Yes

2. Has the statistical analysis been performed appropriately and rigorously?

Reviewer #1: N/A

Reviewer #2: N/A

3. Have the authors made all data underlying the findings in their manuscript fully available?

Reviewer #1: No

Reviewer #2: No

4. Is the manuscript presented in an intelligible fashion and written in standard English?

Reviewer #1: Yes

Reviewer #2: Yes

Reviewer #1: I have two major concerns that need to be addressed before this manuscript can be considered for publication, as well as several additional suggestions for improvement.

1. Methodological Concerns – Interpretive Phenomenology

The description of the analysis process aligns more closely with thematic analysis than with the principles of interpretive phenomenology (IP). One of the core tenets of IP is its idiographic commitment, which typically involves a detailed, in-depth examination of a small, homogenous sample. In the current manuscript, it is unclear how these foundational principles were implemented.

Moreover, the Results section reads primarily as descriptive rather than interpretive. I encourage the authors to consult exemplar IP studies to better understand how to move beyond surface-level description toward deeper interpretive engagement with participants’ lived experiences.

2. Conceptual Clarity – Use of the Term "Paradox"

The manuscript frequently employs the term paradox to describe participants’ dilemmas or experiential tensions. However, a paradox is generally understood as a situation or statement that appears self-contradictory or illogical, but which may hold a deeper truth upon closer examination. Most of the examples presented reflect internal conflicts, tensions, or identity struggles—not paradoxes in the strict sense. I recommend that the authors either define the term more clearly or consider replacing it with more appropriate concepts such as conflict, ambivalence, or tension.

3. Additional Comments

The lack of alignment between the research questions and the interview guide should be explicitly acknowledged and explained in the Method section.

Including sample interview questions would enhance transparency and clarify how data were elicited.

The statement that the researchers aimed “not to influence participants’ answers” (p. 9, line 243) reflects a positivist stance that does not align with the ontological and epistemological assumptions of interpretive research. I encourage the authors to reconsider this framing in light of their chosen paradigm.

The Discussion section could be more concise and focused. Reducing its length and sharpening its analytical insights would improve its overall impact.

Addressing these issues would substantially strengthen the conceptual, methodological, and analytical coherence of the manuscript.

Reviewer #2: Abstract

•The abstract presents the objective, research context, and methodology, and includes three main findings. However, the findings are presented as general formulations ("experience full of paradoxes," "multitude of experiences, very different from what had been imagined," "finding their own way to make sense of their experiences") rather than in a way that characterizes the unique insights of the research.

•The clinical and research implications are mentioned in overly general terms. It could be more effective to include one concrete example of a clinical implication (e.g., the importance of acknowledging paradoxes as part of grieving processes) to illustrate the practical value of the findings.

Introduction

•Despite grief being the central concept, there is only partial reference to this concept, with a lack of reference to grief in general and not only in the context of COVID-19. The introduction focuses primarily on the potential effects of COVID-19 restrictions on bereaved individuals, but lacks presentation of basic theoretical background on grief processes in general. There is no definition of what grief is, how it typically manifests, or what the central theories in the field are. In the absence of this "baseline," it is difficult for readers unfamiliar with the content domain to understand the full significance of the disruptions caused by COVID-19 in this area.

Method

•The study mentions that the analysis was conducted with the involvement of additional researchers to ensure reliability (triangulation), but lacks a detailed description of the methodological process for handling disagreements between researchers. It is unclear how discussions were conducted when there were different interpretations of the same data, or what the criteria for decision-making were. This information is vital for assessing the reliability of the findings and methodological transparency.

Findings

•There is a notable imbalance in length between themes: Theme 2 spans 155 lines compared to 88 lines in Theme 1, creating a disproportionate presentation.

•There is repetition of key ideas - for example, the need for physical and social contact is mentioned at least three times in the same basic descriptive manner, and the sense of unreality recurs four times in identical contexts. While repetition of findings in qualitative research can strengthen their reliability and emphasize their importance, each repeated mention should add a new layer, deepen understanding, reveal a different angle, or connect to other themes.

Discussion

•The discussion ends mid-thought without a comprehensive summary or clear conclusions, leaving a sense of incompleteness.

•There is a mismatch between the limited sample (12 participants) and the general claims in the discussion, where the authors use language such as "people experienced multiple paradoxes" as if representing the entire population, raising significant questions about generalizability and requiring more explicit acknowledgment of research limitations.

Clinical and Research Implications

•The clinical recommendations are too general and lack concrete guidelines - for example, how exactly to "identify paradoxes" or "explore emotions arising from meaning-making" in actual practice.

•The new concept "grieving for the ideal goodbye" is mentioned as a clinical implication but lacks detail on how to identify it in treatment and what specific interventions are recommended for this condition.

•The research implications are too broad ("include mediators," "person-centered approaches") and need more specific elaboration regarding which mediators and which methodologies exactly are recommended.

**Do you want your identity to be public for this peer review?** For information about this choice, including consent withdrawal, please see our Privacy Policy

Reviewer #1: No

Reviewer #2: **Yes: ** Dr. Yael Doft

---

## [Author Response · Author response to Decision Letter 1]

14 Aug 2025

All answers to reviewers' and editors' comments are listed in the “Response to Reviewers” file.

Dear academic editor and dear reviewers,

First of all, we would like to thank you sincerely for your constructive feedback on the manuscript entitled “What does it mean to have experienced the death of a relative in a context of social and funeral restrictions? Lessons from the pandemic for bereavement research and clinical practice” and submitted to Plos One in April 2025.

We have taken note of all your comments, which have given us food for thought. They have led us to rework our analyses, to deepen and clarify our interpretation, and, we hope, to improve the study and its manuscript. You will find our responses to all your comments below. Our answers are in italics and immediately follow the comment to which they are addressed. We have included our thought process, as well as indications of the changes we have made to the text. We hope you find our work relevant, and that it meets your expectations.

NB: all indications of modifications in the manuscript refer to the document entitled "Manuscript", not to the one containing the tracked changes.

Comments from the editor

The reviewers have completed their assessment of your manuscript, and while they found it engaging, they raised several significant concerns that need to be addressed. I concur with the reviewers, particularly regarding methodological issues. Specifically, your description of the analytic process appears more consistent with thematic analysis than with interpretative phenomenology (IP), and your Results section tends toward description rather than interpretation. Additionally, I recommend you carefully reconsider the conceptual clarity around your use of the term "paradox," as it is frequently employed to characterize participants' dilemmas or experiential tensions.

We agree with all the points raised and have reworked our analyses and the manuscript as much as possible to follow your and the reviewers' recommendations, thus improving the quality of the study and manuscript. You will find our answers to the specific points of methodology and conceptual clarity around the term "paradox" below.

We also made changes to the manuscript to ensure compliance with Plos One's guidelines regarding titles, tables and references.

Comments from Reviewer 1 :

I have two major concerns that need to be addressed before this manuscript can be considered for publication, as well as several additional suggestions for improvement.

1. Methodological Concerns – Interpretive Phenomenology

- The description of the analysis process aligns more closely with thematic analysis than with the principles of interpretive phenomenology (IP). One of the core tenets of IP is its idiographic commitment, which typically involves a detailed, in-depth examination of a small, homogenous sample. In the current manuscript, it is unclear how these foundational principles were implemented.

Moreover, the Results section reads primarily as descriptive rather than interpretive. I encourage the authors to consult exemplar IP studies to better understand how to move beyond surface-level description toward deeper interpretive engagement with participants’ lived experiences.

Answer: Thank you for your insightful comment, which gave us food for thought.

We understand the importance of clarifying our analysis in light of your comments, as we recognize that thematic analysis and phenomenological analysis are two distinct approaches. We understand that, in some respects, the description of our analysis method could correspond to thematic analysis. However, this does not reflect the work carried out, the idiographic approach taken throughout the analysis, or our deep involvement (particularly that of the principal investigator) in analyzing each singular experience in detail and in-depth before searching for common or divergent meanings. This was not adequately represented in our description of the method. Consequently, we have revised this section (pp.12-13, lines 285-310) to describe our method using a more appropriate vocabulary and more precise elements that reflect our serious use of the method of interpretative phenomenological analysis. We hope that these modifications meet your expectations.

We also recognize that our study contains a relatively large sample size for the IPA method, and we have included that limitation in the text (p.41, lines 997-1001). However, while this does not preclude in-depth and interpretative analysis, it does require rigor in several respects (Smith, 2011), which we attempted to guarantee throughout the analysis. Firstly, during the analysis, we were careful to take the time necessary to immerse ourselves in each interview in a thorough and idiographic manner, analyzing each extract individually to ensure that our analysis reflected the lived experience of all participants. Secondly, we were careful to reflect the density of each theme in the results, by providing enough extracts to transparently illustrate each theme, as recommended for large samples (Smith, 2011). Thirdly, since the available dataset was large, we focused on analysis on the meaning of loss and grief experienced within the context of restrictions. We did not code or interpret aspects of the participants’ experience unrelated to this specific question. Thus, the experience studied was common, and the sample was relatively homogeneous, as they all lived a loss in a restrictive context, although the ways of experiencing it were very diverse.

That said, after further reflection, we agree that the results section was overly descriptive and did not sufficiently meet the IPA's requirements for depth and interpretation. Consequently, we revisited the interviews and re-read the set of meanings to gain a more interpretive perspective. This led us to modify the meaning and formulation of certain results. For example, we refined the paradoxes as experiences of paradoxical injunctions, and we developed the third theme to capture the meanings made and not just the processes through which people constructed them. Overall, we expanded our analyses and provided more interpretive insights in the results section.

2. Conceptual Clarity – Use of the Term "Paradox"

- The manuscript frequently employs the term paradox to describe participants’ dilemmas or experiential tensions. However, a paradox is generally understood as a situation or statement that appears self-contradictory or illogical, but which may hold a deeper truth upon closer examination. Most of the examples presented reflect internal conflicts, tensions, or identity struggles—not paradoxes in the strict sense. I recommend that the authors either define the term more clearly or consider replacing it with more appropriate concepts such as conflict, ambivalence, or tension.

Answer: Thank you for this enlightening comment. Thanks to your previous recommendation to rework the results, we were able to deepen our understanding of people's experiences and in particular regarding the concept of paradox.

The notions of ambivalence and conflict do appear in our results at various points, and they are appropriate for describing some participants' experiences. However, we believe that they are insufficient, and that the notion of paradox remains a relevant, complementary angle for understanding aspects that are not captured by the notion of conflict, such as experiences of nonsense, absurdity, and illogic.

To clarify this point, we took our analysis a step further and refined our interpretation. As a result, we found that the notion of paradoxical injunctions offers a more precise and relevant insight into the participants’ experiences. Paradoxical injunctions, as developed by Bateson (Bourocher, 2019; Watzlawick et al., 2014), are the imposition of a double constraint, made up of obligatory and mutually exclusive elements, that are therefore impossible to satisfy. The contradictions between the fields of the pandemic and death and mourning seem to have acted as paradoxical injunctions, forcing some participants to reconcile these two incompatible fields: "find humanity in technicalized and sanitized rites", "honor your dead but quickly", "perform social rites but without getting together". This concept appears particularly relevant for various reasons.

On the one hand, the concept encompasses the idea of imposed injunctions and participants being forced to respond to them. Whether these injunctions are internal or external, explicit or implicit, or based on internalized norms, social or cultural learning, or ethical or moral principles, they can be experienced as obligations. This specificity is relevant given the feeling of being forced, the impression of not having had the choice, and the powerlessness experienced by some participants.

On the other hand, paradoxical injunctions allow us to capture the various emotional experiences reported by the participants. First, they can produce feelings of fear and guilt (Bourocher, 2019), which are shown in our results. Second, they can also create internal conflicts, ambivalence, or emotional dissonance when they are internalized, as seen in Tania’s account. Third, paradoxical injunctions can produce experiences of paradox, by combining discordant elements simultaneously (Nardone & Portelli, 2007), creating feelings of absurdity, and nonsense (Bourocher, 2019). In our results, people who had to combine these irreconcilable requests experienced paradoxical rituals, which were made up of elements that could not make sense together and gave rise to feelings of absurdity (p.17, lines 385-392), illogic (p.18, lines 425-431), and meaninglessness (p19, lines 447-450, p.20, lines 474-476). Finally, paradoxical injunctions can paralyze people, as they are “caught between two impossible demands” (Bourocher, 2019, p. 366). This feeling of stupefaction and difficulty of movement was also shared in our results (p.25, lines 570-578).

In conclusion, we believe that the concept of paradoxical injunctions is appropriate to capture comprehensively some participants’ experiences because it encompasses the notion of obligation, the potential internal conflicts, emotional dissonance and guilt that could result, as well as the experience of paradox and the resulting sense of nonsense. This concept was brought in the results (pp.20-21, lines 481-486), further developed in the discussion (pp.33-34, lines 806-821) and in the clinical implications (pp.37-38, lines 917-924).

3. Additional Comments

- The lack of alignment between the research questions and the interview guide should be explicitly acknowledged and explained in the Method section. Including sample interview questions would enhance transparency and clarify how data were elicited.

Answer: It is true that this element was only mentioned in the limitations section and therefore lacked transparency. Following your recommendation, we mentioned it in the Method section (p.11, lines 258-262) and provided further explanation (p.15, lines 341-353). For greater transparency, we also added sample questions and follow-up examples (pp.11-12, lines 262-272).

- The statement that the researchers aimed “not to influence participants’ answers” (p. 9, line 243) reflects a positivist stance that does not align with the ontological and epistemological assumptions of interpretive research. I encourage the authors to reconsider this framing in light of their chosen paradigm.

Answer: You are absolutely right; thank you for that comment. That sentence was intended to demonstrate how the wording of the questions aimed at allowing diverse narratives and unexpected elements to emerge. It has been rewarded to align more with our epistemological position (p.11, line 258).

- The Discussion section could be more concise and focused. Reducing its length and sharpening its analytical insights would improve its overall impact.

Answer: Thank you for this suggestion. We had clarified our messages and reduced the length of the discussion. We decided to focus our discussion on the more original contributions: the paradoxical injunctions, the grief for the ideal goodbye, the evolving and complex meanings. We decided to leave some less essential elements of discussion, such as types of meaning-making processes. We hope it will appear clearer and more impacting.

References

Bourocher, J. Injonction paradoxale. [paradoxical injunction] In Vandevelde-Rougale A, Fugier P, editors. Dictionnaire de Sociologie Clinique. Eres;2019. pp.365 367.

Smith JA. Evaluating the contribution of interpretative phenomenological analysis. Health Psychology Review. 2011 Mar;5(1):9 27. Doi : 10.1080/17437199.2010.510659.

Nardone G, Portelli C. Caught in the middle of a double‐bind: the application of non‐ordinary logic to therapy. Silvia Broecker M, éditeur. Kybernetes. 2007 Aug 14;36(7/8):926 31. Doi : 10.1108/03684920710777432.

Watzlawick P, Beavin JH, Jackson DD, Morche J. Une logique de la communication. [A logic of communication]. Paris: Points; 2014.

Comments from Reviewer 2:

1. Abstract

- The abstract presents the objective, research context, and methodology, and includes three main findings. However, the findings are presented as general formulations ("experience full of paradoxes," "multitude of experiences, very different from what had been imagined," "finding their own way to make sense of their experiences") rather than in a way that characterizes the unique insights of the research.

Answer: You are exactly right. Thank you for highlighting this point. Following your recommendation, we further expanded on the main findings and explained them more precisely to emphasize the originality of the study's unique contributions (you will notice that we modified some of our results to be more interpretive, as suggested by another comment):

“In our findings, the context of the pandemic appeared to be completely incompatible with the field of dying and mourning, creating paradoxical injunctions for some participants. While this led to feelings of guilt, powerlessness and loss of meaning for some, others were able to experience meaningful moments and to find solace in the farewell. All experiences were far from what had been expected, for better or for worse, and participants had to find their own ways to make sense of these unexpected experiences. The meanings they gave were complex, combining different levels of meaning - personal, moral, societal, or existential - and evolved over time, as did their emotional experiences.” (p.2, lines 35-43).

- The clinical and research implications are mentioned in overly general terms. It could be more effective to include one concrete example of a clinical implication (e.g., the importance of acknowledging paradoxes as part of grieving processes) to illustrate the practical value of the findings.

Answer: You are right. Again, we have expanded the abstract by specifying some clinical and research implications to highlight the practical value of our study:

“The results highlight the notions of paradoxical injunctions and grieving for an ideal goodbye as relevant for understanding and supporting the bereaved, drawing avenues for therapeutic work (e.g., restoring agency by providing secure space to live a ritual). They also have implications for research, highlighting the need to broaden the understanding of “impact”, to include mediators assessing subjectivity, and to privilege person-centered qualitative and quantitative methods.” (p.2, lines 43-48).

2. Introduction

- Despite grief being the central concept, there is only partial reference to this concept, with a lack of reference to grief in general and not only in the context of COVID-19. The introduction focuses primarily on the potential effects of COVID-19 restrictions on bereaved individuals, but lacks presentation of basic theoretical background on grief processes in general. There is no definition of what grief is, how it typically manifests, or what the central theories in the field are. In the absence of this "baseline," it is difficult for readers unfamiliar with the content domain to understand the full significance of the disruptions caused by COVID-19 in this area.

Answer: Thank you for your well-founded comm

---

## [Editor Report · Decision Letter 1]

25 Aug 2025

What does it mean to have experienced the death of a relative in a context of social and funeral restrictions? Lessons from the pandemic for bereavement research and clinical practice.

PONE-D-25-18637R1

Dear Dr. Boever,

We’re pleased to inform you that your manuscript has been judged scientifically suitable for publication and will be formally accepted for publication once it meets all outstanding technical requirements.

Kind regards,

Michal Mahat-Shamir, Ph.D.

Academic Editor

PLOS ONE

Additional Editor Comments (optional):

I have carefully reviewed the revisions and amendments made by the authors to the manuscript, as well as their responses to the reviewers’ comments. I am impressed by the thorough and comprehensive work that has been done, particularly in aligning the writing with the principles of IPA. I greatly appreciate the effort invested and find the manuscript, in its current version, suitable for publication.
---

## [Editor Report · Acceptance letter]

PONE-D-25-18637R1

PLOS ONE

Dear Dr. Boever,

I'm pleased to inform you that your manuscript has been deemed suitable for publication in PLOS ONE. Congratulations! Your manuscript is now being handed over to our production team.

Kind regards,

on behalf of

Prof. Michal Mahat-Shamir

Academic Editor

PLOS ONE